# STDP-Driven Rewiring in Spiking Neural Networks under Stimulus-Induced and Spontaneous Activity

**DOI:** 10.3390/biomimetics8030320

**Published:** 2023-07-20

**Authors:** Sergey A. Lobov, Ekaterina S. Berdnikova, Alexey I. Zharinov, Dmitry P. Kurganov, Victor B. Kazantsev

**Affiliations:** 1Laboratory of Neurobiomorphic Technologies, The Moscow Institute of Physics and Technology, 117303 Moscow, Russia; kazantsev@neuro.nnov.ru; 2Neurotechnology Department, Lobachevsky State University of Nizhny Novgorod, 603022 Nizhny Novgorod, Russia; berdnikova-k@mail.ru (E.S.B.); zharinov@neuro.nnov.ru (A.I.Z.); 3Laboratory of Neuromodeling, Samara State Medical University, 443079 Samara, Russia; dmi21v@gmail.com

**Keywords:** spiking neural network, synaptic plasticity, structural plasticity, rewiring, learning, wiring vector field, weight vector field, activity vector field, STDP

## Abstract

Mathematical and computer simulation of learning in living neural networks have typically focused on changes in the efficiency of synaptic connections represented by synaptic weights in the models. Synaptic plasticity is believed to be the cellular basis for learning and memory. In spiking neural networks composed of dynamical spiking units, a biologically relevant learning rule is based on the so-called spike-timing-dependent plasticity or STDP. However, experimental data suggest that synaptic plasticity is only a part of brain circuit plasticity, which also includes homeostatic and structural plasticity. A model of structural plasticity proposed in this study is based on the activity-dependent appearance and disappearance of synaptic connections. The results of the research indicate that such adaptive rewiring enables the consolidation of the effects of STDP in response to a local external stimulation of a neural network. Subsequently, a vector field approach is used to demonstrate the successive “recording” of spike paths in both functional connectome and synaptic connectome, and finally in the anatomical connectome of the network. Moreover, the findings suggest that the adaptive rewiring could stabilize network dynamics over time in the context of activity patterns’ reproducibility. A universal measure of such reproducibility introduced in this article is based on similarity between time-consequent patterns of the special vector fields characterizing both functional and anatomical connectomes.

## 1. Introduction

Researchers in the field of neural network modelling have generally concentrated on changes in the efficiency of synaptic connections represented in models by synaptic weights (see, e.g., [1]). This is synaptic plasticity. In neuroscience, it is believed to be the key cellular basis of learning and memory in animals and humans. 

In artificial neural networks based on formal neurons (ANNs), Hebbian plasticity, which depends on the correlated activity of connected neurons, can be considered as a biologically relevant learning rule [2,3]. In spiking neural networks (SNNs) composed of dynamical spiking units, a biologically relevant learning rule is based on so-called spike-timing-dependent plasticity or STDP [4]. However, in living brain circuits, there are several forms of activity-dependent (adaptive) changes in network structure and functioning. Specifically, neuronal plasticity not only includes synaptic modifications but also homeostatic plasticity [5], structural plasticity [6,7] and other modifications, caused, for example, by astrocyte modulations [8,9,10,11]. Structural plasticity includes the formation of new connections (network growth), the removal of “unnecessary” synapses (pruning), and rewiring synaptic connections without changing their number.

In simulations, it is possible to operate with a fully connected network and express the effects of both synaptic and structural plasticity through changes in the weight matrix ***W***. However, despite the likely coexistence of these two types of plasticity, biological and computational considerations require that the “algorithms” for changing synaptic efficiency and synaptic connectivity should be expressed explicitly. For example, in most areas of the brain, including the mammalian cerebral cortex, only a small fraction of all possible connections between neurons physically exist, even within a local area [12]. In such sparse networks, the possibility of structural plasticity can substantially increase the number of functionally distinct circuits available for encoding learning information [13]. Theoretical analysis showed that overall learning-related memory capacity is maximized in the presence of both synaptic and structural plasticity [13,14,15].

Under experimental conditions, structural plasticity can be registered in cases of a sufficiently strong external influence. Numerous studies report the formation or removal of synapses in the somatosensory cortex after behavioral enrichment [16,17] and sensory stimulation [18,19]. Differential rearing affects dendritic branching in certain areas of the cerebral cortex such as the visual cortex and hippocampus [20]. Experimental protocols involving sensory periphery lesions are also widely used. For example, research shows that digit [21] or limb [22] amputation results in the massive reorganization of cortical chains and axonal growth. A similar rearrangement has also been observed in the visual cortex after retinal injury. After a few months, the cortical region corresponding to the lesion locus becomes sensitive to the perception from the intact part [23,24,25]. 

However, in neuroscience and medicine, the role of structural plasticity in precisely tuned processes such as learning and memory is still debated. In this aspect, simulation models of spiking neural networks with structural plasticity could serve as a useful tool for the estimation of the role of different plasticity forms in the function’s formation. 

In modeling, structural plasticity is typically not self-sustained but complements homeostatic or synaptic plasticity. Specifically, in the proposed models with homeostatic plasticity, the formation and removal of new connections depends on the activity of neurons, which can be calculated directly through the firing rate [26] or indirectly through the simulated calcium concentration in neurons [27,28]. Long-term synaptic plasticity is usually represented by STDP or its modifications [29,30,31]. In such cases, old synaptic connections are removed based on a threshold, meaning connections with weights below a certain value are deleted. New synaptic connections usually form between randomly selected neurons, although some studies also take into account the distance between neurons. Thus, the probability of a new connection decreases with increasing distance between neurons [27,32,33]. Multicomponent models that incorporate multiple types of plasticity—short-term and long-term synaptic plasticity, homeostatic rules, and structural plasticity—have also been utilized in various studies [33,34].

In particular, these models have been instrumental in verifying experimental results, such as the restoration of network activity during “sensory deprivation” [27]. An SNN with rewiring exhibits cortex-like structural features that cannot be random [33]. Moreover, modeling studies have predicted the important role of structural plasticity in the formation of network functionality, learning, and memory processes. Specifically, adaptive rewiring has been shown to form neuronal assembles [29] and network architectures with different target connectivity patterns [32,35]. Structural plasticity with multisynaptic connections can stabilize neural activity and connectivity [30]. Rewiring and synaptic pruning can improve learning both in biophysics [34] and computational SNNs that incorporate the properties of deep artificial neural networks [36,37,38]. Finally, [26] demonstrates the possibility of implementing associative learning based on structural plasticity. 

This paper describes a model of the spiking neural network equipped with STDP and structural plasticity capable of reconfiguring the network connectome in an activity-dependent manner. This adaptive rewiring enables the consolidation of the effects of STDP in response to a local external stimulation of the SNN. The work consistently develops the vector field approach for monitoring the dynamics of the functional, synaptic, and anatomical connectome. Finally, a vector-based measure of activity patterns’ reproducibility is introduced which allows one to reveal the network stabilization effect of the adaptive rewiring. 

The structure of the rest of the paper is as follows. “Section 2. SNN model” describes the neuron dynamics and the synapse model. Then, the description of the developed model is presented in “Section 3. The model of Synaptic Plasticity with Rewiring”. The vector field methods are given in Section 4. The results of computational experiments are described in subsections “Section 5.1. Network rewiring under stimulus-induced activity” and “Section 5.2. Rewiring and stability of neural network during spontaneous activity”. Finally, in Discussion and Conclusions, a generalization of the results is made, the limitations of the model are discussed, and assumptions about future research are made.

## 2. SNN Model

### 2.1. Spiking Neuron

The approach described previously [39,40] was employed to simulate the dynamics of the SNN. Specifically, the dynamics of a neuron were described by Izhikevich’s model [41]. In terms of functionality, this model exhibits similarities to the Hodgkin–Huxley model, but it demands significantly fewer computational resources, making it particularly suitable for simulating large-scale neural networks [42]. The subsequent dynamical system provides a description of this model:(1)$$\frac{dv}{dt}=0.04v^{2}+5v+140-u+I(t),$$
(2)$$\frac{du}{dt}=a(bv-u),$$
with the additional condition to reset the variables when the spike peak is reached:(3)$$\mathrm{if}\;v\geq +30\;\mathrm{mV},\;\mathrm{then}\;\left\{\begin{array}{c} v\leftarrow c\\ u\leftarrow u+d^{\prime } \end{array}\right.$$
where v is the transmembrane potential, u is the recovery variable, a,b,c,d are the parameters, and I(t) is the external current. When the potential v reaches a threshold of 30 mV, a spike is recorded, and v and u are reset to the values specified in Equation (3). In the work, the following parameters were used: a=0.02;b=0.2;c=−65; and d=8. In the absence of an external input, these values allow the neuron to remain in a resting state. However, the presence of an external current results in regular spiking, which is typical for cortical “RS” neurons [41,42]. In Equation (1), the external current was presented by the following:(4)$$I(t)=\xi (t)+I_{syn}(t)+I_{stml}(t)$$
where ξ(t), is uncorrelated Gaussian white noise with zero mean and standard deviation D = 5.5, $I_{syn}(t)$ is the synaptic current. The term $I_{stml}(t)$ represents the stimulation current. For stimulated neurons (see Section 5.1), the external stimulus is delivered as a sequence of pulses with a frequency of 10 Hz, duration of 3 ms, and amplitude sufficient to excite the neuron. 

### 2.2. Synaptic Model and Network Connectivity

The synaptic current was calculated by taking the weighted sum of the output signals from neurons connected to a specific postsynaptic neuron i:(5)$${I_{i}}_{syn}(t)=\sum_{j}g_{j}w_{ij}y_{ij}(t),$$
where $g_{j}$ represents the transformation coefficient, which converts the output signal of the presynaptic neuron *j* into a synaptic current (g=20 arb. units for excitatory and g=−20 arb. units for inhibitory neurons), $w_{ij}$ is the weight of the synaptic connection, and $y_{ij}(t)$ is the output (or synaptic) signal from neuron *j* to *i*, which corresponds to the neurotransmitter released at synapses with each pulse. The dynamics of the transmitter can be described via the Tsodyks–Markram model, which accounts for short-term synaptic plasticity [43]: (6)$$\frac{dx_{ij}}{dt}=\frac{z_{ij}}{\tau_{rec}}-u_{ij}^{*}x_{ij}\delta (t-t_{j}-\tau_{ij}),$$
(7)$$\frac{dy_{ij}}{dt}=-\frac{y_{ij}}{\tau_{I}}+u_{ij}^{*}x_{ij}\delta (t-t_{j}-\tau_{ij}),$$
(8)$$\frac{dz_{ij}}{dt}=\frac{y_{ij}}{\tau_{I}}-\frac{z_{ij}}{\tau_{rec}},$$
(9)$$\frac{du_{ij}^{*}}{dt}=\frac{u_{ij}^{*}}{\tau_{facil}}+0.5(1-u_{ij}^{*})\delta (t-t_{j}-\tau_{ij}),$$
where $x_{ij},y_{ij},z_{ij}$ are the fractions of the neurotransmitter in a restored, active, and inactivated state; $t_{j}$ is the time of the presynaptic spike, determined via Equation (3); $\tau_{I}$, $\tau_{rec}$, and $\tau_{facil}$ are the characteristic times of the processes of inactivation, restoring, and facilitation; $\tau_{ij}$ is the axonal delay in the arrival of a spike to the synaptic terminal; and $u_{ij}^{*}$ is the part of the neurotransmitter released from the restored fraction $x_{ij}$ at each spike. In the work, such parameter values were chosen that made it possible to demonstrate both the effects of synaptic depression (in the case of high-frequency activity) and synaptic facilitation (in the case of activity with a frequency of about 1 Hz): $\tau_{I}$ = 10 ms, $\tau_{rec}$ = 50 ms, and $\tau_{facil}$ = 1000 ms. Axonal delays were proportional to the distances between neurons (see below).

Both simple 1D and 2D architectures of the SNNs were considered. In the 2D case, the analysis was conducted for the organized (structured) connectome and a more realistic case when neurons (up to 500 units) were distributed in a planar layer at random position sites. In the latter case, the units were randomly coupled with the probability of interneuron connections decreasing with the distance, according to the Gaussian distribution:(10)$$f=\frac{1}{\sqrt{2\pi }\sigma }e^{-\frac{d^{2}}{2\sigma^{2}}},$$
where σ is the standard deviation chosen to obtain an average length of synaptic connections *d* at 50 μm. This architecture captures the essential features of in vitro neuronal cultures, allowing for the reproduction of their dynamic modes, e.g., network bursting [44,45,46].

## 3. The Model of Synaptic Plasticity with Rewiring

Long-term synaptic plasticity was represented by STDP. The STDP was simulated using the algorithm with local variables, described in [47]:(11)$$\frac{ds_{i}}{dt}=-\frac{s_{i}}{\tau }+\delta (t-t_{i})$$
(12)$$\frac{ds_{j}}{dt}=-\frac{s_{j}}{\tau }+\delta (t-t_{j}-\tau_{ij})$$
(13)$$\frac{dw_{ij}}{dt}=F_{+}(w_{ij})s_{j}(t)\delta (t-t_{i})-F_{-}(w_{ij})s_{i}(t)\delta (t-t_{j}-\tau_{ij}),$$
where $s_{i}$ and $s_{j}$ are variables that track spikes on the postsynaptic and presynaptic neurons, respectively, τ = 10 ms is the characteristic decay time of local variables, and $t_{i}$ and $t_{j}$ are the spike generation time on the postsynaptic (receiving spikes) and presynaptic (transmitting spikes) neuron. In turn, the weight increase and decay functions follow the multiplicative rule [47,48]:(14)$$F_{+}(w_{ij})=\lambda (1-w_{ij})$$
(15)$$F_{-}(w_{ij})=\lambda \alpha w_{ij},$$
where λ=0.001 is the learning rate, and α=5 is the asymmetry parameter that determines the ratio of the processes of the weakening and strengthening of the synapse.

The model of structural plasticity included the following SNN-rewiring algorithm. Each 1 s simulation time weight of all of the synaptic connections was checked. The pruning process removed all of the connections with weights smaller than the threshold value $w_{min}$ for a duration of $t_{w}$ (“lifetime of weak connection”). Whenever a connection was removed, a new connection with weight $w_{new}$ (“weight of newborn connections”) was created. In this step, a postsynaptic neuron was randomly selected, whereas the choice of the presynaptic neuron was determined via the distance according to (10). 

Figure 1 presents an example of rewiring in a simple SNN. The weight of the connection from neuron 8 to neuron 4 ($w_{48}$) fell below the threshold value $w_{min}$ over a period $t_{w}$. Thus, this connection was deleted. Then, postsynaptic neuron 2 was randomly selected for the newborn connection. The choice of the presynaptic neuron was determined via the distance, according to (10). For example, the probability was higher for neuron 5 compared to the other neurons. The weights of the newborn connection $w_{25}$ were set as $w_{new}$. 

The formation of new connections in the model had two options. In the first option, only a single synaptic connection between two neurons was allowed. In this case, if a connection from the chosen presynaptic neuron had already been established, the process of searching for a candidate connection was repeated. In the second option of multiple connections, this check was not performed, and the number of synaptic connections between a pair of neurons was not limited.

## 4. Vector Fields for Visualizing Functional and Structural Rearrangements in an SNN

To describe functional and structural rearrangements in an SNN on a large scale, the approach of the vector field suggested earlier in [40,49] required an expansion. The square substrate covered by the network was divided into an N×N grid. A possible connection going from neurons j to i, located at positions $p_{j}$ and $p_{i}$ ($p_{i}$, $p_{j}\in R^{2}$), was represented as a connection vector directed from the presynaptic to the postsynaptic neuron: (16)$$c_{ij}=l_{ij}\frac{p_{i}-p_{j}}{{\left\|p_{i}-p_{j}\right\|}_{2}}$$
where $l_{ij}$ is the length of the connection vector calculated depending on the type of the vector field (see below).

All vectors passing through cell (k,m) in the grid were added to obtain the resulting vector for the given cell:(17)$$C_{km}=\sum_{ij\in \wedge km}c_{ij}$$
where ∧km is the set of vectors that have a nonempty intersection with cell (k,m). Then, the matrix ($C_{km})\in M_{N_{F}{\times N}_{F}}(R^{2})$ defines the large-scale vector field. 

Three types of vector fields were introduced for this study: (i) the activity vector field reflecting the functional connectome, (ii) the weight vector field reflecting the synaptic connectome, and (iii) the wiring vector field reflecting the anatomical connectome. 

For *the wiring vector field* (Figure 2A), the length of the connection vector was determined via the existence of coupling between neurons
(18)$$l_{ij}=\left\{\begin{array}{c} 1,if\;neuron\;i\;\mathrm{has}\;a\;\mathrm{synaptic}\;\mathrm{connection}\;\mathrm{from}\;j\\ 0\;\mathrm{otherwise} \end{array}\right.$$

For *the weight vector field* (Figure 2B), the length of the connection vector was determined via the weight of the connection between the presynaptic neuron j and the postsynaptic neuron i.
(19)$$l_{ij}=w_{ij}$$

Finally, for *the activity vector field* (Figure 2C), the length of the connection vector was determined via the history of spikes passing through this connection and the resulting excitation of the postsynaptic neurons:(20)$$\frac{dl_{ij}}{dt}=s_{j}\delta \left(t-t_{sp}^{i}\right)-\frac{l_{ij}}{\tau_{l}},$$
where $s_{j}$ is the activity of presynaptic neuron (12), $t_{sp}^{i}$ is the time instant of the postsynaptic spike i, and $\tau_{l}$ is the relaxation time.

Note that since $l_{ij}=l_{ij}(t)$ for all three types of vector fields (in Equation (18) due to rewiring, in Equation (19) due to STDP, and in Equation (20) according to the definition), the vector field (16) can change over time. Having three types of vector field is convenient for estimating the contribution of different types of plasticity to functional and structural rearrangements. In fact, *the weight vector field* is the most informative due to the combination of features from both the wiring vector field (e.g., absence of a connection from neuron *j* to *i* is equivalent to $w_{ij}$ = 0) and the activity vector field (due to STDP rule; also see [40]).

Also note that multidirectional synaptic connections can lead to mutual compensation and eventually to a zero vector even if connections pass through the corresponding cell (e.g., Figure 1A, the bottom left connections). In the case of *the activity vector field*, zero vectors are observed when spikes do not pass through the synaptic connections, thus causing response excitation in the postsynaptic neuron (Figure 2C, the top connection).

In order to numerically integrate the model Equations (1)–(15) and (20), the Euler method was used with a time step of 0.5 ms. This approach has been proven to be suitable for integrating large systems of Izhikevich’s neurons [41,42]. To facilitate this process, a custom software platform called NeuroNet was developed using the cross-platform QT IDE. NeuroNet provides online simulations of the model and the construction of vector fields. The software is written in C++ and can perform real-time simulations of SNNs with tens of neurons using an Intel^®^ CoreTM i3 processor. One can access NeuroNet at [50].

## 5. Results

### 5.1. Network Rewiring under Stimulus-Induced Activity

First, let us consider a simple 1D architecture of the SNN equipped with STDP and structural plasticity (Figure 3). The assumption here is that, in the initial state, neurons have bidirectional synaptic connections with relatively small weights of *w* = 0.3 (Figure 3A). Then, the central neuron is stimulated, which induces the propagation of spikes from the center to the periphery. The activity vector field illustrates these dynamic events in a single static picture in Figure 3B. Consequently, this repetitive spike propagation causes STDP-driven weight rearrangement (the details can be found in [39,40,49]), which is visualized by the weight vector field (Figure 3C). In turn, structural plasticity replaces unused centripetal connections with multiple centrifugal ones involved in long-term activity, as illustrated in Figure 3D. Thus, this stereotypical stimulation leads to network rearrangements at different levels: spikes → synaptic weights → anatomical connectome, visualized in the static pictures of Figure 3 by the vector fields.

Figure 4 represents a more complicated case of a 2D neural circuit with radial (centrifugal and centripetal relative to stimulus location) and ring (tangential) connections. Similar to the 1D case, the stimulus-induced effects can be observed. First, centrifugal spiking activity appears (Figure 4B), leading to the potentiation of centrifugal connections and to the depression of other (centripetal and tangential) connections (Figure 4C). Rewiring captures these functional rearrangements into structural changes in the SNN (Figure 4D).

Figure 5 illustrates stimulus-induced changes in a medium-scale (200 neurons) neural network mediated by STDP and rewiring. Note that immediately after the appearance of local stimulation (Figure 5, “*t* = 1 s”), only the activity field (Figure 5A,B) changes, while the weight field (Figure 5C) or the wiring field (Figure 5D) remain unaffected. In this case, the activity field vectors reflect the general direction of spike activity in the form of traveling waves or propagating patches of activity.

However, over time (Figure 5, “*t* = 100 s”), due to STDP, the weights of synaptic connections in the network change, enhancing the conduction of the traveling waves induced by the stimulus. This effect is achieved due to the strengthening of centrifugal connections (relative to the place of stimulation), as illustrated by the vectors of the synaptic weight field (Figure 5C), which are oriented in the direction from the place of stimulation (see details in [40]).

After some time (Figure 5, “*t* = 1000 s”), the rewiring reinforces the STDP effects, resulting in predominantly centrifugal connections in the network structure. This is illustrated by the wiring vector field (Figure 5D), which largely coincides at time *t* = 1000 s with the weight field (Figure 5C) and the activity field (Figure 5B). Thus, the vector fields show the consistent potentiation of pathways for the predominant conduction of spike activity, first by changing the efficiency (weights) of synaptic connections and then by changing the structure of the network.

### 5.2. Rewiring and Stability of Neural Network during Spontaneous Activity 

Over time, spontaneous (arising from neural noise (4)) spiking activity leads to rearrangements of the connectome through STDP and rewiring. In contrast to the stimulus-induced activity, these changes occur much more slowly and in an irregular manner. The anatomical and functional connectomes of the network gradually change, which is reflected in the changes in the vector fields. 

To estimate the degree of changes in the SNN over time, the following quantity is introduced. For all vectors of the weight field $C_{km}^{I}$ at time $t_{1}$ and the vectors $C_{km}^{II}$ at time $t_{2}$, the cosine similarity was calculated as follows:(21)$$s_{km}^{I–II}=\frac{\left(C_{km}^{I},C_{km}^{II}\right)}{\left\|C_{km}^{I}\right\|\left\|C_{km}^{II}\right\|}$$

Then, the similarity between the vector fields can be calculated as the average cosine similarity of all vectors of the compared fields:(22)$$S^{I–II}=(\sum_{k,m=1}^{N,N}s_{km}^{I–II})/N^{2}$$

Thus, in the limit cases, a similarity value $S^{I–II}=1$ corresponds to complete coincidence of weight vector fields at times $t_{1}$ and $t_{2},$ while $S^{I–II}=-1$ describes totally different fields.

Figure 6A shows an example of weight vector fields at different times during the rewiring-induced network dynamics. The corresponding values of similarity, *S,* for pairs of the vector fields were also calculated. Note that the stabilizing effect of rewiring can be observed after 1000 s of rewiring (the similarities of vector fields at times 0–1000 s are *S*^0–I^ = 0.71, whereas at times 1000–2000 s they are *S*^I–II^ = 0.71 = 0.94). The dynamics of the vector field’s similarity for SNNs with and without rewiring are illustrated in Figure 6B. The functional changes in the network can be observed to converge to a stable pattern in the presence of rewiring. These dynamics can be interpreted as the self-organization of the network, where only a certain pattern of “useful” synaptic connections survive. In the presence of only STDP, the network is more flexible and has enough time to rebuild the functional connectome under the influence of spontaneous activity. Moreover, in this case, one can observe significant drops in the similarity coefficient of vector fields reaching *S* = 0.27, which indicates intense weight rearrangements. This phenomenon can be explained by the presence of the so-called superburst activity observed experimentally in planar neuronal cultures [44].

To assess the effect of rewiring parameters on stability, a neural network was tracked for 15,000 s of spontaneous activity, and its state (connections and weights) was recorded every 1000 s of simulation time. Then, the average value of the similarity coefficient for all 15 networks (paired comparison) obtained was calculated, which was used as a measure of the network stabilization. Figure 7 shows the results of the simulations of neural networks with different parameters of rewiring. A statistically significant connectome stabilization effect is observed for all values of the weight of newborn connections $w_{new}$. At the same time, the SNN with single synaptic connections tends to enhance its structural changes as $w_{new}$ increases (Figure 7A). These results are consistent with [30], showing that multicontact synapses can stabilize a neural network.

Increasing the lifetime of weak connection $t_{w}$ leads to the suppression of the stabilization effect, both in the case of single and multiple connections (Figure 7C). However, statistically significant differences in the connectome stabilization of neural networks with and without rewiring were not revealed, only in the case of a very large value of $t_{w}$ = 5000 s. Thus, the developed approach based on vector fields demonstrates that rewiring under conditions of spontaneous activity leads to the stabilization of functional and anatomical connectomes in a wide range of parameters.

## 6. Discussion

Spiking neural networks have attracted significant attention from researchers and engineers due to the general expectation that the performance of an SNN-based artificial intelligence (AI) system will supersede (approaching “brain performance”) traditional ANN-based AI solutions. As a “more biologically relevant” version, spiking neurons offer greater degrees of freedom for information representation and processing. In particular, information can be encoded in time characteristics of spike discharges, rate, and phase measures, as well as in the characteristics of nonlinear dynamics such as self-oscillation, multistability, and chaos. Such an enhancement in the degree of flexibility of SNNs, in turn, raises new challenges related to the tuning and control of specific dynamical modes associated with learning or information function [51]. 

In mathematical modeling and the engineering design of biologically relevant SNNs, tuning is typically based on the effect of spike-timing-dependent plasticity (STDP). In fact, STDP implements a Hebbian learning rule for neurons with spiking dynamics and induces changes in the synaptic weights. Unlike ANNs composed of “digital” formal neurons, SNN units exhibit ongoing fluctuating signals of membrane potentials, currents, ionic concentrations, and changing synaptic weights. Consequently, the dynamics of the SNN and its outcome functions become less predictable and harder to control. In other words, the dynamical spiking patterns encoding information in SNNs cannot be stable in terms of reproducibility over time. However, many experimental studies conducted under both in vitro and in vivo conditions have demonstrated the precise reproducibility of patterns in living neuronal networks [52,53]. One example of this reproducibility is the repetitive spiking sequences in cortical circuits, known as cortical songs [53]. Moreover, these patterns are believed to be internal representations of sensory information. Multiple dynamical mechanisms, including homeostatic plasticity mediated by the brain extracellular matrix, may also contribute to such stability [54]. 

The present study has led to an unexpected discovery of the fact that structural plasticity, which represents a mechanism of activity-dependent appearance and disappearance of synaptic connections, can lead to the stabilization of SNN dynamics in terms of activity pattern reproducibility. The study also introduces a universal measure of such reproducibility, which is based on similarity in time-consequent patterns of the special vector fields characterizing both functional and anatomical connectomes. The observation of spontaneous SNN dynamics revealed that the structural plasticity of the SNN rewires itself by suppressing connections deemed to be “non-useful” for the current type of activity and generating new “useful” synapses. In this context, combining STDP with structural plasticity “amplifies” the general Hebbian paradigm, optimizing the anatomical architecture of SNNs.

Finally, such optimization, which provides reproducibility, can be helpful in designing new learning strategies for SNN-based AI systems.

## 7. Conclusions

In conclusion, the present study has produced a mathematical model of a spiking neural network enhanced with STDP and structural plasticity. The model incorporates the rewiring of the network connectome by eliminating “non-useful” ones and multiplying “useful” ones. Over time, this activity-dependent rewiring can stabilize network dynamics in the context of activity pattern reproducibility. 

Novel methods of vector fields capable of assessing network dynamic reproducibility in terms of synaptic, functional, and anatomical connectomes have been proposed.

Note also that such dynamic self-reconfiguring of the model opens up several directions of future research in the field of SNN design for AI applications. There are two basic limitations of the presented model framework that have to be further addressed. To reproduce particular brain circuit functions, SNN architecture has to imitate a concrete type of spatial morphology that can be much more complicated than layered patterns of cortical networks. Take, for instance, interacting hippocampal regions, thalamic networks, olivocerebellar systems, and others [55]. Obviously, structural plasticity in such “genetically structured” networks has to obey some additional rules so that the newborn connections can fit the original structure pattern. The second point concerns the extracellular medium. Growing neurites forming new interneuron connections go through this medium guided by the field of active chemicals (growth factor molecules and neurotransmitters) that can affect the formation of both anatomical and functional connectomes. Many recent papers in SNN modeling have already addressed several modulation effects of such fields formed, for example, by astrocytes on SNN dynamics [9]. Moreover, the above-mentioned brain extracellular matrix may also serve as a network growth guide [56]. We believe that the development of the presented model, taking these factors into account, permits one to design a neuro-mimetic SNN capable of reproducing particular functions of living brain networks. 

## Figures and Tables

**Figure 1 biomimetics-08-00320-f001:**
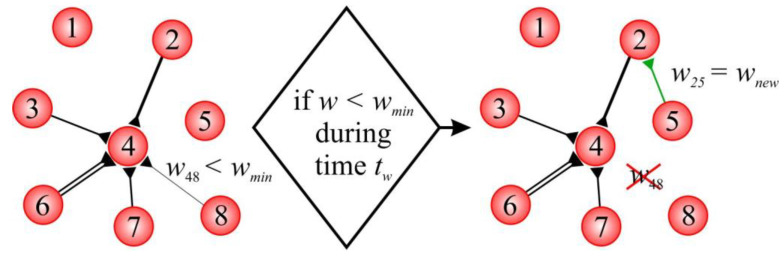
Example of rewiring in the model.

**Figure 2 biomimetics-08-00320-f002:**
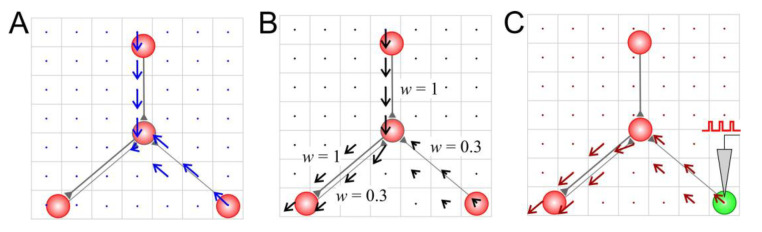
Vector fields of a small neural circuit: (**A**) the wiring vector field, (**B**) the weight vector field, and (**C**) the activity vector field after 20 s stimulation of a neuron (marked in green).

**Figure 3 biomimetics-08-00320-f003:**
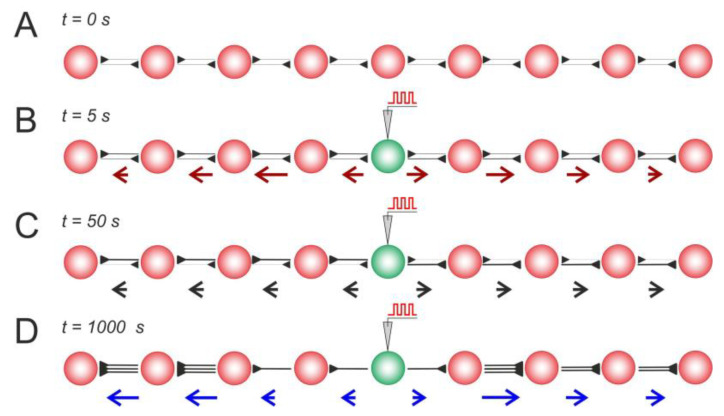
Stimulus-induced changes in a neural line circuit mediated by STDP and structural plasticity: (**A**) the initial condition, (**B**) the activity vector field reflects stimulus-induced activity, (**C**) STDP-driven weight rearrangement, and (**D**) rewiring mediated by STDP and structural plasticity. As in Figure 2, the red/black/blue arrows represent the activity/weights/wiring vector field. The rewiring parameters were as follows: $w_{min}$ = 0.05, $t_{w}$ = 5 s, and $w_{new}$ = 0.1.

**Figure 4 biomimetics-08-00320-f004:**
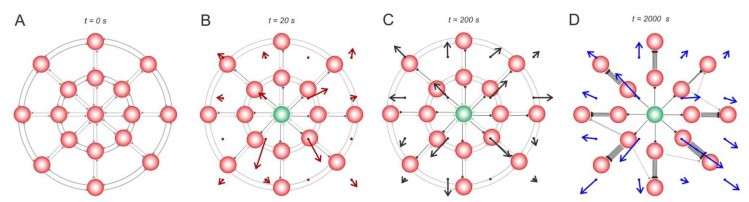
Stimulus-induced changes in a 2D neural circuit. (**A**) the wiring vector field, (**B**) the weight vector field, and (**C**) the activity vector field after 20 s stimulation of a neuron, (**D**) rewiring captures these functional rearrangements into structural changes in the SNN. The rewiring parameters were set as follows: $w_{min}$ = 0.2, $t_{w}$ = 5 s, and $w_{new}$ = 0.05.

**Figure 5 biomimetics-08-00320-f005:**
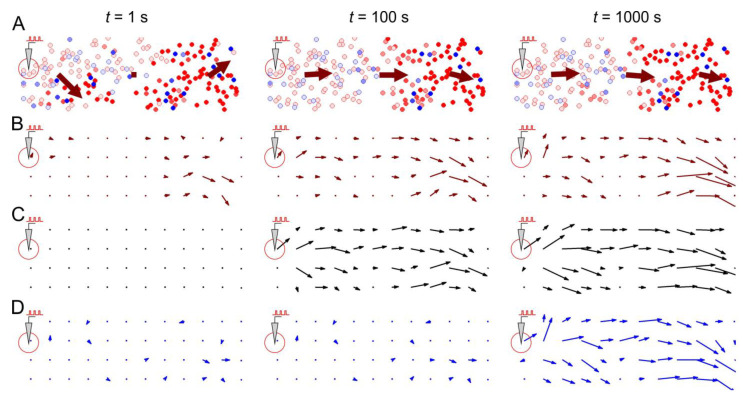
Stimulus-induced changes in a neural network mediated by STDP and structural plasticity: (**A**) general view of the neural network and propagating spike activity at different time points after the start of the stimulation, (**B**) the activity vector field, (**C**) the weight vector field, and (**D**) the wiring vector field. Rewiring parameters were set as follows: $w_{min}$ = 0.05, $t_{w}$ = 100 s, and $w_{new}$ = 0.1.

**Figure 6 biomimetics-08-00320-f006:**
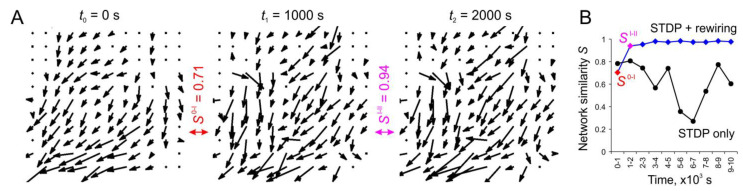
The effect of SNN stabilization under structural plasticity: (**A**) example of weight vector fields at different time points after the introduction of rewiring and the similarities of vector fields *S*; (**B**) dynamics of the vector fields’ similarity in the case of structural plasticity (STDP + rewiring) and without it (STDP only).

**Figure 7 biomimetics-08-00320-f007:**
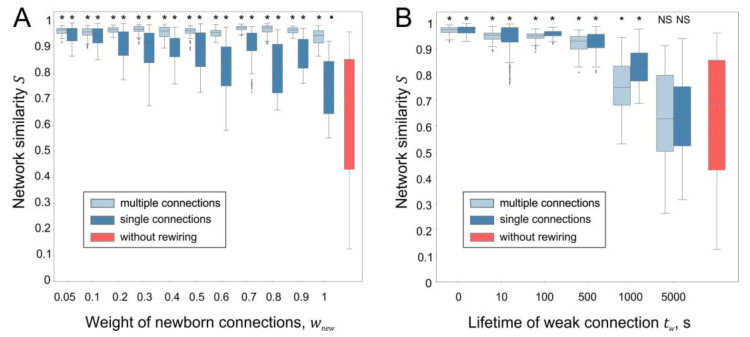
Influence of rewiring parameters on the stabilization of the neural network: (**A**) network similarity vs. weight of newborn connections ($w_{min}$ = 0.05, $t_{w}$ = 5 s); (**B**) network similarity vs. lifetime of weak connections ($w_{min}$ = 0.05, $w_{new}$ = 0.06). *, ●—statistically significant differences between the current network with rewiring and the network without rewiring (*—*p* < 10^−7^, ●—*p* < 0.01, Mann–Whitney U test).

## Data Availability

Not applicable.

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
