# Peer review of "STDP-Driven Rewiring in Spiking Neural Networks under Stimulus-Induced and Spontaneous Activity"

_biomimetics, 2023, doi:10.3390/biomimetics8030320_

Round 1

Reviewer 1 Report

1. Abstract is too long, I would recommend reducing it. Additionally, try to avoid using acronyms in the abstract.

2. Avoid using we/our throughout the paper.

3. List the main contributions clearly in the introduction.

4. Provide paper structure at the end of the introduction.

5. Make sure that each parameter in every equation has been explained in the text.

6. Elaborate more the possible practical applications, to justify the proposed approach.

7. Also, discuss about the limitations in more details.

8. Finally, the conclusion is very limited. It should wind up the research, mention limitations and indicate possible directions of the future research.

Author Response

We thank the reviewer for comments helping to improve the manuscript. For convenience, all new parts of the text in MS have been marked blue.

  1. Abstract is too long, I would recommend reducing it. Additionally, try to avoid using acronyms in the abstract.

We revised the abstract tacking into account this recommendations.

  1. Avoid using we/our throughout the paper.

We revised the MS according this comment and made corresponding corrections.

  1. List the main contributions clearly in the introduction.

We have listed the contributions in the penultimate paragraph of the introduction.

  1. Provide paper structure at the end of the introduction.

Done.

  1. Make sure that each parameter in every equation has been explained in the text.

We revised the MS according this comments and added the descriptions of all parameters (or sets of parameters) and provided all values of parameters (see notations of Figures for variable parameters).

  1. Elaborate more the possible practical applications, to justify the proposed approach.

We developed a custom software called NeuroNet which is available at http://spneuro.net and referred to it in the MS. The basic SNN architectures described in the paper are available on the site also (http://spneuro.net/downloads/nets/set_rewiring.zip). Further we plan to develop and improve our neurosimulator in constant interaction with neuroscience community.

  1. Also, discuss about the limitations in more details.

We see two basic limitations of the presented model framework that has to be further addressed.

1. To reproduce particular brain circuit functions the SNN architecture has to imitate a concrete type of spatial morphology. Obviously, that structural plasticity in such “genetically structured” networks has to obey some additional rules so that the newborn connections must fit the original structure pattern.

2. The second point concerns the extracellular medium. Growing neurites forming new interneuron connections go through this medium guided by the field of active chemicals (growth factors molecules, neurotransmitters) that can affect the formation of both anatomical and functional connectomes.

We believe that the development of the presented model with accounts of these factors permits to design neuro-mimetic SNN capable to reproduce particular functions of living brain networks. 

  1. Finally, the conclusion is very limited. It should wind up the research, mention limitations and indicate possible directions of the future research.

We extended the conclusion taking into account this comment.

Reviewer 2 Report

The problem of neural networks, biologic and artificial, is an important one and very difficult to study and to model. This paper presents interesting results and discoveries.

What doesn’r result clearly: how was the proposed model simulated? How were the experiments done? The methods are not enough described. Mathematical and theoretical, as what kind of simulations using which environment? Is this an independent study or has it a long term aim? Will it be used further for biology, in mathematics, in circuits? It's just that probably, depending on where the results are used, they should be subsequently "moulded" in one way or another.

In terms of data testing results, some tables should be welcome, to synthesize the weight and parameter values  in all the situations, to give the reader an overview of the work performance.

Bibliography properly refered, thoroughly ordered; very well explained the referred parts for sustaining the present work. On one hand the references show the long term preocupation of the authors for this subject but on the other hand, a little too many autoreferences…

In Figure 1 (line 178), probably the title “Example of rewiring in the model” would suffice, and the whole explanation would be better contained in the text with referrance to Figure1, instead of containing the explanation in the figure title.

In line 59: theore-tical  

In line 108: instead “elsewhere” maybe a concrete indication would be more appropriate

In 199 I’d recommend a blank line for a better readability of the formula (18) of the connection vector length

Author Response

We thank the reviewer for comments helping to improve the manuscript. For convenience, all new parts of the text in MS have been marked blue.

The problem of neural networks, biologic and artificial, is an important one and very difficult to study and to model. This paper presents interesting results and discoveries.

What doesn’r result clearly: how was the proposed model simulated? How were the experiments done? The methods are not enough described. Mathematical and theoretical, as what kind of simulations using which environment? Is this an independent study or has it a long term aim? Will it be used further for biology, in mathematics, in circuits? It's just that probably, depending on where the results are used, they should be subsequently "moulded" in one way or another.

We extended the description of the methods in the paper. Specifically, in order to numerically integrate the model equations, the Euler method was used with a time step of 0.5 ms. This approach has been proven to be suitable for integrating large systems of Izhikevich's neurons. To implement the calculation we used our software platform called NeuroNet which is available at http://spneuro.net. We have referred to it in the MS now. The basic SNN architectures described in the paper are available on the site also (http://spneuro.net/downloads/nets/set_rewiring.zip). Further we plan to develop and improve our neurosimulator in constant interaction with neuroscience community.

Our approach is to implement the basic principles of brain learning and functioning in SNNs in order to further using in AI applications. In the conclusion section of the paper we discussed limitations and perspective of the model in these context.

In terms of data testing results, some tables should be welcome, to synthesize the weight and parameter values  in all the situations, to give the reader an overview of the work performance.

We didn’t test biology experimental data, since at present, on the one hand, it is impossible to finely assess the anatomical and functional connectome of a living neural network, and, on the other hand, to selectively influence the parameters of structural plasticity. Thus the measuring of the performance of the model is impossible now. As for our numerical simulation study we provided all values of parameters. Specifically the constant values were given in the main text just after describing of each parameter.  The values which we varied we pointed in descriptions of the figures illustrating described phenomena. Additionally as we mentioned the basic SNN architectures described in the paper are available on the site (http://spneuro.net/downloads/nets/set_rewiring.zip) and the SNN files already have parameters value which we used.

Bibliography properly refered, thoroughly ordered; very well explained the referred parts for sustaining the present worsk. On one hand the references show the long term preocupation of the authors for this subject but on the other hand, a little too many autoreferences…

We reduced the number of autoreferences.

In Figure 1 (line 178), probably the title “Example of rewiring in the model” would suffice, and the whole explanation would be better contained in the text with referrance to Figure1, instead of containing the explanation in the figure title.

We followed this suggestion

In line 59: theore-tical 

Corrected

In line 108: instead “elsewhere” maybe a concrete indication would be more appropriate

Corrected

In 199 I’d recommend a blank line for a better readability of the formula (18) of the connection vector length

Corrected